# Incidence and Epidemiology of Citrus Viroids in Greece: Role of Host and Cultivar in Epidemiological Characteristics

**DOI:** 10.3390/v15030605

**Published:** 2023-02-22

**Authors:** Matthaios M. Mathioudakis, Nikolaos Tektonidis, Antonia Karagianni, Louiza Mikalef, Pedro Gómez, Beata Hasiów-Jaroszewska

**Affiliations:** 1Plant Pathology Laboratory, Institute of Olive Tree, Subtropical Crops & Viticulture, ELGO-DIMITRA, Karamanlis Ave. 167, Gr-73134 Chania, Greece; 2Departamento de Biología del Estrés y Patología Vegetal, Centro de Edafología y Biología Aplicada del Segura (CEBAS)-CSIC, P.O. Box 164, 30100 Murcia, Spain; 3Department of Virology and Bacteriology, Institute of Plant Protection—National Research Institute, ul. Wł. Wegorka 20, 60-318 Poznan, Poland

**Keywords:** *Citrus* sp., citrus viroids, viroids detection, viroids frequency, distribution and epidemiology, mixed infections

## Abstract

Viroids represent a threat to the citrus industry and also display an intricate matter for citrus tristeza virus (CTV) control as most of the commercial citrus rootstocks that are resistant/tolerant to CTV appear to be highly susceptible to viroid infection. Therefore, a detailed knowledge of the viroid’s incidence and distribution, along with the assessment of unexplored epidemiological factors leading to their occurrence, are necessary to further improve control measures. Herein, a large-scale epidemiological study of citrus viroids in five districts, 38 locations and 145 fields in Greece is presented, based on the analysis of 3005 samples collected from 29 cultivars of six citrus species. We monitored the occurrence of citrus exocortis (CEVd), hop stunt (HSVd), citrus dwarfing (CDVd), citrus bark cracking (CBCVd), and citrus bent leaf (CBLVd) viroids, and addressed their epidemiological patterns and factors shaping their population structure. Our results show a high frequency and wide distribution of four viroids in all areas and in almost all hosts, whereas CBLVd occurrence was restricted to Crete. Mixed infections were found in all districts in which a wide spread of viroids was observed. We identified a potential pathogens’ different preferences that could be partially explained by the host and cultivar, including the type of infection (single or mixed) and the number of viroids in the mixed infections. Overall, this work provides the first detailed epidemiological study on citrus viroids, enriching our knowledge for the implementation, production, and distribution of certified citrus propagative material, and the development of sustainable control strategies.

## 1. Introduction

Citriculture (*Citrus* sp.) is one of the most important crops worldwide, with an annual production of 145 million tons [1], providing a significant source of income especially to the third world countries. The Mediterranean diet is universally accepted as a factor in well-being and longevity, and the importance of citrus fruits as a cornerstone in this philosophy of the human food chain is particularly valued, as citrus fruits provide a rich source for nutrients and antioxidants. However, citriculture is threatened by the occurrence of phytopathological diseases, especially those caused by viruses and viroids, degrading tree yield factors and final product quality, and increasing inputs [2], therefore, knowledge of their distribution and epidemiology is necessary to sustain healthy crops and to develop control strategies.

In Greece, citriculture covers more than 415 hundred hectares, with more than 19 million trees, consisting of the third most important fruit crop, after olive and other fruit trees (pome-stone fruits), which indicates its immense social-economic importance to the country [3]. Citruses are mainly grown in southern and northwestern parts of Greece with Crete Island, Peloponnese, and Arta covering 80% of the cultivation. Oranges (*Citrus sinensis*), mandarins (*C. reticulata*) and lemons (*C. limon*) account for the most cultivated citrus species and production, whereas grapefruits (*C. paradisi*), blood oranges, limes (*C. latifolia*, *C. aurantifolia*) and other *Citrus* sp. (e.g., bergamot, citron, kumquat) are less cultivated and with a lesser contribution to the annual production.

The cultivation of citrus fruits in Greece has been in crisis in recent years, even though they occupy the first place in fruit exports and being an important pillar in its agricultural production. New plantings are carried out in areas where citrus fruits are grown, while others are replaced, resulting from disease losses and the use of alternative *Citrus* species, following global market trends with the aim of reducing imports/inputs, but without following a national framework to prevent the insertion and spread of diseases. One of the most important citrus diseases is citrus tristeza virus (CTV), causing huge economic losses of millions of euros that changed the citrus industry worldwide [4]. Globally, the management of CTV is based mainly on the use of resistant rootstocks, free of viroids to which they are susceptible. CTV is a quarantine pathogen and 20 years after the first report in Greece of the disease in orange trees in the Peloponnese and Crete [5], it has recently taken the form of endemics in various regions and outbreaks alongside the emergence of new strains and invasion of new hosts [6,7,8]. In Greece, the application of sour orange as the main rootstock, which is known for its high CTV susceptibility, together with the limited studies on the viroids presence, minimizes chances of a successful application of control strategies, such as the use of resistant rootstocks combined with the knowledge of viroids incidence and epidemiology studies.

To date, eight viroid species assigned to four different genera of the family *Pospiviroidae* infecting citrus have been reported: citrus exocortis viroid (CEVd) of the genus *Pospiviroid*, hop stunt viroid (HSVd) of *Hostuviroid*, citrus bark cracking viroid (CBCVd) of *Cocadviroid*, and citrus dwarfing viroid (CDVd), citrus bent leaf viroid (CBLVd), citrus viroid V (CVd-V), citrus viroid VI (CVd-VI), citrus viroid VII (CVd-VII) of *Apscaviroid* [9,10,11,12]. They can be transmitted by vegetative propagation, grafting, cuttings and pruning tools causing various symptoms like bark cracking, dwarfing, stunting, leaf bent and yellowing [13]. The citrus viroids are single-stranded RNA molecules of 279–372 nucleotides [9,13]. Although citrus viroids have been detected in many countries, wide range epidemiology studies and the impact of host and/or cultivar susceptibility/infectivity or tolerance remain insufficient to gain in depth information on viroids’ biological characteristics which could be used in the development of new control strategies. In Greece, a small survey at the National Germplasm Collection has revealed the presence of five viroids in five different species and recently the detection of viroids in lime was reported [14,15].

In this study, we perform a large-scale national survey, monitoring the occurrence of CEVd, CDVd, CBCVd, HSVd and CBLVd infecting citrus species in the three main citrus-growing regions of Greece: Crete (three provinces), Peloponnese (one province) and Arta. We investigated the frequency and the epidemiological factors that may affect the distribution of the five viroids in six different citrus species and different cultivars. The results reported herein, represent a milestone in the knowledge of citrus viroids’ frequency and epidemiology, to assess and understand potential factors that could impact their epidemiology and population structure.

## 2. Materials and Methods

### 2.1. Field Surveys, Citrus Hosts, Cultivars and Plant Tissue Sampling

A total of four independent surveys were carried out during the period between November 2020 to June 2022, corresponding to the citrus growing periods of autumn (September–October; two conducted surveys, one in 2020 and one in 2021) and spring (April–June; two conducted surveys, one in 2021 and one in 2022), as the most suitable detection periods of citrus virological pathogens. A total of five provinces (different geographical districts) of Greece, which represent the main citrus-growing production areas, were selected: Chania, Heraklio and Rethimno from the Crete region, Argolida from the Peloponnese region and Arta from the Ipiros region (Figure 1; lower panel). In total, 145 fields scattered along the citrus provinces, corresponding to 38 different locations (Figure 1) of the country were one-time surveyed in order to assess the detection, frequency, and epidemiological characteristics of CEVd, CDVd, CBCVd, HSVd and CBLVd in Greece.

We surveyed six different citrus hosts: orange, lemon, mandarin, grapefruit, lime, and blood orange. Some of the species were either not able to be found in some districts (due to climate conditions or soil demands they are not cultivated) or were surveyed in a lower number due to limited available fields (Appendix A). From each species, and depending on the host, up to 11 different cultivars, either endemic or foreign were examined for the presence of viroids. A total of eleven different cultivars were collected from orange, six from lemon, five from mandarin, three from grapefruit and blood orange, and one from lime (Appendix A). For each survey field symptoms, if any, were inspected and registered. The number of the collected samples was proportional to the abundance of the host in each field and from different spots along the field, ensuring a representative sampling procedure.

All samples were one-time collected during the first three surveys, whereas the sampling of spring 2022 was focused to re-collect samples which needed to be evaluated again for PCR confirmation. A total of 3005 individual citrus trees samples (Appendix A) were randomly collected from both symptomatic and asymptomatic trees, within or in the immediate vicinity of different fields. As most of the citrus viroids during the first years of infection do not cause any visible symptoms, four fresh twigs (10–20 cm) of the annual vegetation with four to five leaves per tree, from the four quarters of the plants were collected, regardless of the symptoms’ presence to avoid any bias.

### 2.2. Viroid Detection

A mixture of plant tissues consisting of bark, leaves and petioles were used from each quarter of the sampled trees and ground into a fine powder with liquid nitrogen. Total RNAs were isolated from 0.1 g ground tissue by the TriZol method, as previously described [16]. The quality of the RNAs was validated by electrophoresis of the samples in agarose gels under denaturing conditions, and the concentration was measured spectrophotometrically using the Q5000 UV-Vis spectrophotometer (Quawell, CA, USA). Each sample was diluted to a final concentration of 130 ng/μL. The total RNAs were further used as a template in a one-step RT–PCR assay using viroids-specific primers [17,18]. A slightly modified common assay profile was used for the detection of viroids. Briefly, the 25 µL RT–PCR reaction mix contained Green-Go Taq Flexi buffer (Promega, Madison, WI, USA), 3 mM MgCl_2_, 5 mM DTT, 0.4 mM dNTPs, 0.4 µM of each primer, 10 U RNase Inhibitor (NEB, Hitchin, England), 1.25 U MML-V (Minotech, Crete, Greece), 1.5 U Go-Taq polymerase (Promega), and 1.5 µL of total RNAs, under the following cycling scheme: 50 °C for 60 min, 95 °C for 10 min, 40 cycles of 94 °C for 30 s, 60 °C for 30 s, 72 °C for 60 s and a final step of 72 °C for 7 min. The size of the amplified RT–PCR products is expected to vary from 286 to 371 bp, depending on the viroid-primer pair used. Moreover, a two-step RT–PCR was performed in the cases where the results of the initial assays were not clear (low band intensity, generation of unspecific products) and needed confirmation. The synthesis of cDNA was according to the manufacturer’s instructions and the detection protocol was as previously described [18].

### 2.3. Cloning, Sequence Analysis and Variability

An isolate from each geographical district and from different hosts was chosen to confirm the viroid origin (CEVd: mandarin, Argolida; HSVd: blood orange, Arta; CDVd: orange, Rethimno; CBCVd: grapefruit, Heraklio; CBLVd: lemon, Chania). The RT–PCR amplicons amplified from the citrus viroids’ detection were purified using a column gel extraction system (Macherey-Nagel, Düren, Germany) and cloned into a pGEM-T Easy® cloning vector (Promega), according to the manufacturer’s instructions. Positive clones were Sanger sequenced in both orientations using the universal M13 forward and reverse primers by Macrogen (Amsterdam, The Netherlands). Nucleotide sequences were compared using the Blast-n software. To estimate the divergence levels, multiple nucleotide alignments were performed using Clustal-X (Conway Institute UCD, Dublin, Ireland) [19] along with the available sequences from GenBank.

### 2.4. Data Analyses and Processing

The obtained data from the field surveys were processed and analyzed using R 4.2.2 software [20] and the R-studio environment [21]. For bar charts the “ggplot2” package was used. The size proportional circles relative to the number of infected samples were created by the “bubblemap” package, in which three different colors representing three different numerical sample groups (1–10, 11–80, 81–300) were used due to the wide range of infected samples obtained between the studied locations. For the mixed viroid infection types the “treemap” package was used to visualize their frequency during the survey periods. Different formulas were used to perform the data statistical processing (expressed in percentages). An example is the following which was used to calculate the frequency of each mixed infection type per host in each area.
$$F=\frac{number\;of\;one\;type\;mixed\;infected\;samples\;per\;host\;in\;each\;area}{number\;of\;mixed\;infected\;samples\;per\;host\;population\;in\;each\;area}\times 100$$

All data were extracted from the Appendix A and in Appendix A all the formulas which were used for the analysis are presented with examples.

## 3. Results

### 3.1. Incidence and Distribution of Citrus Viroids in Greece

During this study all five viroids infecting *Citrus* were detected among the different districts in Greece using the viroid-specific molecular tools. The frequency of the citrus viroids studied herein was estimated from the 3005 samples collected during four surveys in three years, from five geographical districts in 38 locations and 145 field plots (Figure 1). Overall, in the total hosts and pathogens in the country, viroids were detected in a high proportion of the samples (70.18%; 2109 out of 3005 samples). Mixed infections were found at 80.61% (1700 out of 2109) and only 19.39% (409 out of 2109), were singly infected (Figure 2a). Similar frequency rates of infected samples were observed when we analyzed the total number of sample hosts and pathogens in each collection period (autumn, spring), confirming, as already known, the suitability of both periods for citrus viroids’ detection. HSVd was the predominant viroid in Greece (63.69%), and CDVd was the second most prevalent viroid (57%) (Figure 2b). CBCVd and CEVd were also shown in high frequencies as the third and fourth most prevalent viroids (36.57% and 27.25%, respectively), whereas CBLVd was the least detected viroid in the analyzed samples (2.76%).

The results showed a wide distribution of citrus viroids in Greece as they were detected in all five geographical districts, following the same overall pattern presented above (minor variation in frequencies), with similar rates. HSVd was the most predominant, between 51.2 and 84% in all districts, followed by CDVd (45.2–64%), except in the Argolida district in which both were similar detected (Figure 3). CBCVd was the third detected viroid (36.6–37%) in Argolida, Chania and Heraklio followed by CEVd (20.8–31.2%), whereas in Rethimno both viroids were detected in the same number of samples (50%). Only in the Arta area did CEVd have a short, but higher detection frequency (29.9%) compared to CBCVd (25.2%). CBLVd was detected only on Crete Island, in all three provinces, but in lower rates, in contrast to the other citrus viroids. It is worth noting that the higher numbers of infected samples in Rethimno may be attributed to the reduced number (*n* = 50) of collected samples compared to the other districts.

In each of the surveyed districts, the overall percentages of infected samples were high and similar, from 66 to 84%, with the Argolida area showing a slight reduction. An interesting finding was the moderate percentages (9–25%) of singly infected plants and the common characteristic in all districts of the high presence of mixed infections (Appendix A).

### 3.2. Epidemiological Characteristics of Citrus Viroids

In total, oranges were the most surveyed host (34.61%) followed by mandarins and lemons (22.36 and 22.13%, respectively), whereas the other hosts were collected at a lower scale (Appendix A). The present detailed survey in five districts and different locations, revealed a wide spread of citrus viroids, as they were detected in almost all the surveyed locations and all different fields (Figure 1). The overall results showed the presence of at least one viroid infection in all the different hosts sampled, from all the different geographical districts (Table 1). Single infections were present in low percentages and varied between the districts. In the Rethimno area, single infections were detected only for HSVd and most of them were in mandarins (Appendix A). In Argolida, the CEVd was detected only in mixed infections and single CBCVd and CDVd isolates were found only in lemons and blood oranges, respectively, whereas oranges had the higher proportion of single infections mostly with CDVd (33.33%) and limes with HSVd (25%). In Arta, grapefruits were always mixed infections, CBCVd was detected only in mixed infections and lemons showed single infections only for CDVd, whereas HSVd (23.07%) was mostly detected in single infections in mandarins and CEVd only appeared in single isolates in oranges. In the Heraklio district, limes were only in mixed infections, CEVd was detected only in mixed infections, and from all the CBCVd infected samples single isolates were only found in oranges, whereas mandarins had a predominance of single infections, mostly with HSVd (29.76%). Likewise in Chania, CEVd single isolates were found only in lemons, the blood oranges and mandarins were the most detected hosts with single infections (HSVd, CDVd, CBCVd), and HSVd was highly detected in single infections in grapefruits (24.15%). In all areas a common characteristic was the detection of CBLVd only in mixed infections, and the high percentage of mixed infections in all hosts (>65%).

Differences in the epidemiology and frequency of the viroids were shown between the districts and hosts. The CBLVd viroid was predominant in lemons (10.27%) and less in grapefruits (0.98%) in Chania, whereas it was not detected in limes and blood oranges in any of the areas of Crete. HSVd and CDVd were the first and second most predominant viroids in all six host species (in Arta and Argolida for mandarin-lemon and orange-lemon-blood orange-lime, respectively, it was the contrary), except the case of grapefruits in the Heraklio area in which CBCVd was the second most detected viroid (60%) (Appendix A). In Rethimno, CEVd and CBCVd were the next most frequent viroids (40–52.5%) and CBLVd was only detected in oranges. In Argolida, the oranges were the least infected with CEVd (20%) and lemons the most infected (42.5%), whereas the mandarins had the highest infections for CBCVd (41.67%). In limes, all viroids were in high percentages, probably due to the limited number collected. In Arta, the lemons shared the highest infections of CEVd and CBCVd (52.94% and 47.06%, respectively), whereas the oranges and mandarins had the lowest infection frequencies. Likewise, CEVd was the third most prevalent viroid in Heraklio, with lemons having the highest rates of infection (50%) and mandarins the lowest (4.76%), and was not detected in grapefruits. On the contrary, in oranges CBCVd, the fourth most prevalent viroid, was detected in 42.5% of the samples, whereas lemons were the least infected (26.92%). CBLVd was only detected in oranges and mostly in mandarins (1.39%). In the last and most surveyed area of Chania with 2237 collected samples, CBLVd was the less predominant viroid detected in grapefruits, oranges, mandarins and mostly in lemons. Lemons and blood oranges appeared with the highest infection rates for CEVd and CBCVd (52.39% and 44.6%, and 33.78% and 41.89%, respectively), whereas the mandarins and grapefruits showed the lowest infections (8.28% and 24.4%, and 8.5% and 28.43%, respectively). The present study also includes the first report in Greece of CBLVd in lemon, blood orange, mandarin and in some cultivars of orange and grapefruit, CBCVd in blood orange, mandarin, grapefruit and in some cultivars of lemon and orange, CEVd in mandarin, grapefruit, blood orange and in some cultivars of orange and lemon, HSVd in some cultivars from all hosts and CDVd in mandarin and in some other cultivars from the other hosts.

Another diverse characteristic between the districts was the frequency of mixed infection types in hosts. In Rethimno, the oranges were mostly quadruple infected (50%), whereas in mandarins the same frequency of mixed infections with two, three or four viroids was observed (Figure 4). In the district of Argolida, all hosts showed a high frequency of quadruple infections, whereas in limes there were no double infections recorded. In Arta 53.19% of oranges were double infected, lemon samples were mostly detected with three viroids and grapefruits were found mostly with four viroids (66.66%) and no double infections, whereas in mandarins the double and quadruple types were equally detected with no triple infections. The limes in Heraklio were found with four viroids (66.66%) and no samples with triple infection, and the opposite case was for grapefruits, in oranges there was an equal frequency (33.88%) of triple and quadruple infections, whereas in lemons and mandarins the predominant type was the triple infection. In the Chania district the quadruple infection was the most predominant in lemons and especially in blood oranges (68.57%), whereas in limes, oranges and grapefruits there was triple infection, and the mandarins were mainly found with the presence of two viroids (44.56%).

The most predominant combination of double infections in all districts and hosts was the HSVd+CDVd, except in oranges in Argolida where the CEVd+CBCVd was the most detected type, and in Rethimno in mandarins this combination was not found. In Arta, the oranges were detected with three more combinations of two viroids, whereas in Heraklio and Chania the oranges and lemons were found with all different combinations (Appendix A). From triple infections the combination of HSVd+CDVd+CBCVd was the most predominant in all hosts in Chania, Heraklio and Argolida, whereas in Arta the combination of CEVd+HSVd+CBCVd (in this district the oranges and lemons were found with one and two more combinations, respectively) was the most detected, whereas none of the mandarins in Rethimno were reported with this combination, and in lemons from Chania and Argolida the triple type of CEVd+HSVd+CDVd was the most detected. Some unique triple combinations with CBLVd were detected in Chania from lemons. Moreover, the quadruple combination of CEVd+HSVd+CDVd+CBLVd was only found in lemons in Chania, whereas no quintuple infection was found in grapefruits.

### 3.3. Potential Correlation between Pathogen Viroid and Host—Cultivar

The analysis of the overall host-pathogen data collected from five different districts in Greece shows a trend of correlation between pathogens’ presence and host. The oranges and lemons were revealed to be highly infected with viroids (76–80%), compared to mandarins and blood oranges which were shown to be less infected (58%) (Figure 5a). Another parameter which was shown to be associated with the host was the presence of single infections. Grapefruits and mandarins shared the highest percentages of single infections (27%), whereas lemons and limes were detected frequently in mixed infections (85–86%) (Figure 5a).

From the perspective of a specific viroid infection correlated with a specific host, it was noticed that lemons (total number from all areas) were the most infected with all five different viroids (43–72%) (Figure 5b; Appendix A), the grapefruits were the least infected with CBLVd, whereas in limes and blood oranges CBLVd was not detected. The oranges were the second most correlated host with high infectivity to HSVd, CDVd and CBCVd, and blood oranges for CEVd.

Taking into account a fairly representative number of collected samples from various cultivars, a possible correlation between cultivar and viroid infectivity resulted, which impacts the host population structure of the citrus viroids. From oranges, the W. navel, N. late, Saloustiana and Mirodato cultivars were the most infected with viroids, whereas cv. Mpotsato was the least infected and CBLVd was mainly detected in cv. N. late, followed by Valencia, N. hall and W. navel (Appendix A). Adamopoulou and Vakalou cultivars were found more frequently with the presence of viroids compared to Interdonato, and CBLVd was predominant in cv. Vakalou. For mandarins, Clementine and Page cultivars were the most infected with viroids and Nova was the least infected, while the CBLVd was only detected in cv. Ancor. The red grapefruits, in which CBLVd was only found, were shown to be more infected compared to white and pink cultivars, and in blood oranges the Kara Kara cultivar was highly infected with viroids compared to Moro.

Overall in the collected samples, double infections were more frequent in mandarins (42.53%) and triple infections in grapefruits (48.95%) (Figure 6). In total, after the most predominant type of infection, which is the triple infection, the quadruple infection type followed where the blood oranges had the highest frequency. Infections with five viroids are more correlated with lemons (6.93%). It is clear from the results of the presented survey conducted in different districts and hosts, that the different frequency of specific mixed infection types is correlated with the host.

### 3.4. Sequence Analysis of Viroid Isolates, Symptoms Description

The majority of the samples collected were asymptomatic. Nevertheless, variable symptoms were registered in some types of hosts and cultivars. In limes and lemons typical symptoms of bark cracking were often seen in addition to leaf yellowing. In the Heraklio area and specifically in a field with mandarin cv. Kino (a local cultivar), 15% of the trees showed stunting symptoms. It was not possible to corelate any of the observed symptoms with a specific viroid, since as the results above indicated, the percentages of single infections were very low.

The sequence analysis confirmed the viroid origin of CEVd (371 nt; mandarin, Argolida), HSVd (302 nt; blood orange, Arta), CBCVd (386 nt; grapefruit, Heraklio), CDVd (294 nt; orange, Rethimno) and CBLVd (318 nt; lemon, Chania), with no variations to their sizes compared to available sequences in GenBank. The sequences of CEVd, CBCVd and CBLVd also represent the first available viroid sequences from these hosts. All sequences shared high levels of nucleotide identities of 96–100% with other available viroid sequences. The sequences used in this study were deposited in GenBank under accession numbers OQ127287–OQ127291.

## 4. Discussion

The annual citrus production, in more than 140 countries, has been steadily increasing from 2010 to 2018, with a slight reduction during 2019–2020, highlighting it as one of the most widely grown fruit crops worldwide. The trends have also significantly risen with oranges (>50% of the world citrus production) to occupy the first place (>40% of the citrus exports) followed by mandarins, lemons, and grapefruits. In Greece (23rd producing country) citrus production over the years shows an increase, and represents an important pillar of the agriculture industry (the first place in fruit exports are oranges, confirmed herein, as they were the most surveyed host), and is threatened by the growing competition of other fruits and food product categories, new market trends, and notably by the presence of pathogen diseases such as viruses and viroids. The high accumulation and emergence of former and new virus-like diseases results from the lack of national certified propagative material. The application of successful control strategies in citrus relies on knowledge of the incidence of virus-like pathogens and their epidemiology. In Greece, information about the presence of viroids in citrus groves is missing, and in the present study we tried to address their epidemiologic patterns from different hosts and areas, and to interpret any correlation between epidemiological factors.

In Greece, all five viroid species targeted in this study have been previously detected [14,15] in various hosts, and as previously reported, in other citrus growing areas in the world [22,23,24]. In addition, herein the first record of CBLVd, CBCVd, CEVd and CDVd is reported in four citrus species. Among the 3005 analyzed samples in this study, seven out of ten were infected with at least one viroid, highlighting the high level of viroids’ incidence in citrus in Greece. These results are in agreement with previous studies conducted in other countries reporting an average infection rate of 78–96% [25,26,27,28,29], although in Australia a very low incidence (5.6%) was recorded [30]. Among them, HSVd was the predominant viroid in Greece (63.69%), followed by CDVd with slightly reduced infection rates, whereas CBCVd and CEVd were found at a lower frequency. These data are consistent with previous reports in the country [14], and similar to other countries where HSVd and CDVd were the predominant viroids [26,27,28,29,30,31,32]. Contrary to Greece, in China, Tunisia, Lao PDR, Costa Rica, Uruguay, Japan, Australia and Colombia CBCVd was either found in low frequencies (3–7%) or it was not detected [26,27,28,30,31,33,34,35], whereas in Italy CEVd was detected in higher percentages than CBCVd (68% vs. 24%) [29], or in Sudan the CBCVd was the predominant viroid (100%) [36]. CBLVd was the least detected viroid (2.76%), confirming a previous national report in which it was detected only on a single field occasion in oranges [14]. The presence of CBLVd has been recorded in few and mostly eastern countries [29,37,38,39,40], principally as the first reports in some hosts, which also note its low occurrence compared to other citrus viroids, whereas in Japan, China and Tunisia the CBLVd was highly detected (27–39%) [26,27,28].

In this work, the citrus viroids assessment showed a wide distribution among citrus species, as they were all detected in the five geographical districts in Greece, except for CLBVd, which was only detected in three provinces of Crete Island. Nevertheless, an interesting finding was the differences occurred in viroids’ distribution patterns in citrus per district, but following similar overall rates (66–84%). The surveyed areas are covered by lowlands. These areas have similar geomorphological/landscape characteristics and are dominated by non-sloping farms. Regarding the climate conditions, there are some differences between southern to middle-northwestern studied areas, which are mainly noted in the amount of annual rainfall, while minor differences are observed in the range of average temperatures. The sampling time in the two northern areas was adjusted to coincide with similar temperature conditions in the southern areas and achieve overall uniformity among the studied areas. The present paper presents the first and detailed large-scale survey of 3005 samples, allowing a trusted and credible prediction to evaluate the distribution of five viroids in five areas. Previous studies conducted in either less surveyed areas and locations (1–3 areas and 20–202 samples) [26,29,31,33,34,39,41,42], or in 8–16 areas but with a low frequency of samples encountered during the surveys (90–217) [28,32], could lead to doubtful interpretations of the status of viroids’ distribution. With regard to the studies in which a larger amount of samples were tested in Taiwan (*n* = 689) and Australia (*n* = 1800), either only two viroids were evaluated or all the collected samples originated from a National Germplasm Collection and fields of one area [25,30].

Regarding the epidemiological characteristics and specifically the implications of hosts, producing-areas, and type of mixed infections, a wide spread of citrus viroids was observed, as they were detected in almost all the surveyed district’s locations and all different fields, with at least one viroid infection in each of the six different hosts. Variations in the epidemiological patterns were recorded among the different producing-area districts, sharing in common the dominance of mixed infections (>65%) which are known to occur in citrus [24,43], and are in agreement with similar or higher percentages reported in other countries [25,26,27,28,31,33]. The single infections varied between the surveyed areas, not only in relation to the frequencies, with Chania possessing the highest (74.08%) and Rethimno the lowest presence (0.98%) of the total single infections, but also to the viroid species and the hosts observed. The most predominant viroids found in single infections in all areas and almost in all citruses were HSVd and CDVd (except Rethimno), whereas CEVd and CBCVd were present only in mixed infections in some areas, and were absent in others. In agreement with several survey studies, with more than one sampled prefecture area, HSVd and CDVd (except Lao PDR in which the CBLVd was the second) were predominant and present in all areas, but the others were in mixed infections [29,31,33,34]. In some of the districts there were also cases in which some hosts were only either mixed infected, like grapefruits in Arta and limes in Heraklio, or single infected (e.g., blood orange in Arta), information which was never reported in other countries in single areas. Likewise, and confirming other studies [27,28,29,31,36,39], CBLVd was present only in mixed infections in the hosts and areas which were detected, while Pagliano et al. [34] found it in single infections in mandarins in Uruguay.

Each district appeared to have different frequencies of viroids in specific hosts forming dissimilar epidemiology patterns. In common, they shared the predominance of HSVd and CDVd in all host species in the first two places, depending on the studied area, whereas the epidemiology of CBLVd was different, with a high frequency in lemons in Chania, in oranges in Rethimno (5%) and in mandarins in Heraklio (1.36%). Unlike in Greece, where the CBLVd was not detected in limes and blood oranges in any of the areas, it was reported in limes in Malaysia, Iran, Pakistan and Lao PDR [33,39,41,42] and in blood oranges in Tunisia and Italy [27,29]. The CBCVd and CEVd viroids hold the next places with their frequency of varying between hosts in the studied areas, as described in the results. In the Arta district, CEVd was the third most predominant viroid in all tested hosts. A common epidemiology characteristic was that CEVd in lemons had higher frequencies in all areas. Similar results were reported only from two countries [28,36], whereas in Lao PDR, China, Costa Rica, Uruguay and Tunisia the frequency of CEVd in specific hosts was higher than CBCVd [26,27,31,33,34]. It is worth noting that the present data cannot be directly compared with previous studies conducted in small scale surveys of areas, numbers of samples and hosts.

The high occurrence of mixed infections which was observed in all the districts around Greece, revealed four types which varied between the areas and the citrus species, creating different epidemiological statuses. In the Argolida and Rethimno areas the quadruple infections were the most abundant in all different hosts, whereas in Arta and Heraklio they were double and in Chania triple infections, respectively. The frequency rate varied depending on the hosts. Infections with five viroids were found in lemons and oranges mostly in the Chania area, which is recorded for the first time in Greece [14]. It has been reported so far only in Sudan and Italy in lemons, oranges but also in mandarins, grapefruits and blood oranges [29,36]. Likewise, two to four viroid types were reported previously in Greece but not in all of the hosts tested herein [14], with double infections appearing to be predominant in the majority of countries [26,30,33,34,39], whereas in Italy and Tunisia it was the triple type, and in Sudan quadruple infections in all the tested hosts [27,29,36]. The present large-scale survey allowed the discovery of six new combinations of double, triple and quadruple infections which are reported for the first time, with two of them found only in lemons in Chania. In line with previous studies [26,34], the HSVd+CDVd combination shared the highest rates of double infections in all areas and hosts, with some exceptions, and was the second most frequent type of mixed infection in Greece (23.47%). This combination was never reported in Iran, Sudan and Lao PDR [33,36,39], whereas it was found in lower frequencies in Australia, Tunisia and Italy [27,29,30]. The less frequent (0.17%) HSVd+CBLVd and CDVd+CBLVd combinations were detected only in mandarins and lemons, with the first only reported in mandarins and lemons in Lao PDR and Tunisia (4.9–9.5%) [27,28,29,30,31,32,33], and the latter never previously recorded. From the triple infection type the HSVd+CBCVd+CDVd combination was the predominant (22.47%), in contrast to other countries which report the CEVd+HSVd+CDVd combination as the predominant mixed infection type [26,27,28,29,30,33,34], similar to Rethimno and Arta areas herein. Moreover, the CEVd+HSVd+CBCVd+CDVd combination of the quadruple type was the most frequent in Greece (31.29%), in most of the hosts and areas, unlike Italy in which it was found in low rates but similar to Sudan in which it was the predominant type [29,36].

The epidemiology of the citrus viroids has been poorly investigated, since most of the studies addressed only the occurrence of two to three viroids and either their detection in a small number of samples and hosts, or in restricted areas. The collective analysis of the overall samples from this in-depth study which presents the largest sample set evaluated so far in citrus viroids species, citrus species, number of districts and within locations and fields, enriched our knowledge and enabled assessment of five potential correlations between host-viroids interactions. The results showed a trend of the pathogens’ infectivity that correlates with host preference, with oranges (34.61% of the total samples) for example, being more sensitive to viroid infections compared to the more tolerant mandarins (22.36% of the total samples). In addition, the citrus species could be correlated with the presence of single or mixed infections of viroids, with grapefruits circulating the highest percentage of single infections, whereas the majority of limes were always mixed infected. In the correlation of viroids’ preference for citrus species, lemons were the most susceptible to all viroid infections, and oranges were the second most correlated citrus species with HSVd and CDVd infectivity, considering that almost 38% of the infections from these viroids were originating in this species. Previous studies in other countries have shown similar indications of possible correlation between viroids’ preferences for specific hosts, but the limited data did not allow concrete conclusions to be drawn [26,27,31,33,34]. Compared to previous studies, in which the evaluated factors were limited [27,29,34], using a larger data set with different cultivars strongly suggests a possible correlation of viroids with specific citrus species’ cultivars as another parameter to understand the forces shaping the viroids population structures. Some local endemic cultivars were shown to be highly sensitive to viroid infections, whereas only cv. Mirodato sweet orange was found to be more tolerant. Given the number of different citrus species and viroids tested from diverse geographical areas, our work has provided evidence that the type of mixed infection is correlated to the citrus species they infect. The host effect on the population structure and epidemiology has been previously documented for CEVd and ascribed to permissive or selective hosts, genetic variability and their pathogenicity [44,45,46,47], and for CDVd to genetic variability and biological properties [48], as well as for other citrus viruses like CTV [49] or other hosts and viruses [50,51,52,53]. The absolute presence of CBLVd in mixed infections, and particularly in types with more than three pathogens (78%), in specific citrus gives us a tempting hint that suggests CBLVd tends to associate positively in co-infections, as was well documented in cases of multi-host viruses in a large-scale survey of wild hosts in Spain [53]. Future research in the molecular characterization (genetic structure, intra- and inter-group variation values, evolutionary relationships, symptom expression by single viroid infections etc) of the obtained Greek isolates from various hosts will give new insights for a better understanding of the impact of these parameters’ described herein.

The vegetative reproduction of citrus trees in addition to the infected budwoods and top-grafting are the main factors which contribute to the high levels of mixed infections [54]. The importance of mixed infections is crucial, as they favor recombination events which play a significant role in viroids evolution [55] through the emergence of new viroids (e.g., CBCVd was created from recombination of CEVd and HSVd) [56] and strains with severe infectivity [57,58]. This may enhance symptom expression, leading to more damage [24] and importantly, expand their host range by the passage of viroids into new hosts in which they can be more catastrophic, similar to the case of CBCVd in hops [56]. Moreover, the presence of mixed infections prevents the study of symptomatology of single viroids in citrus and the evaluation of their importance as a disease. Our findings increase knowledge of citrus viroids’ frequencies and the epidemiological influence of hosts, producing-area, and type of infection in order to understand their epidemiology in citrus crops, and can be used as a source of prophylactic strategies such as the use of less preferable infectious species and cultivars, implementation of certification schemes, and to help with the development of CTV control strategies.

## Figures and Tables

**Figure 1 viruses-15-00605-f001:**
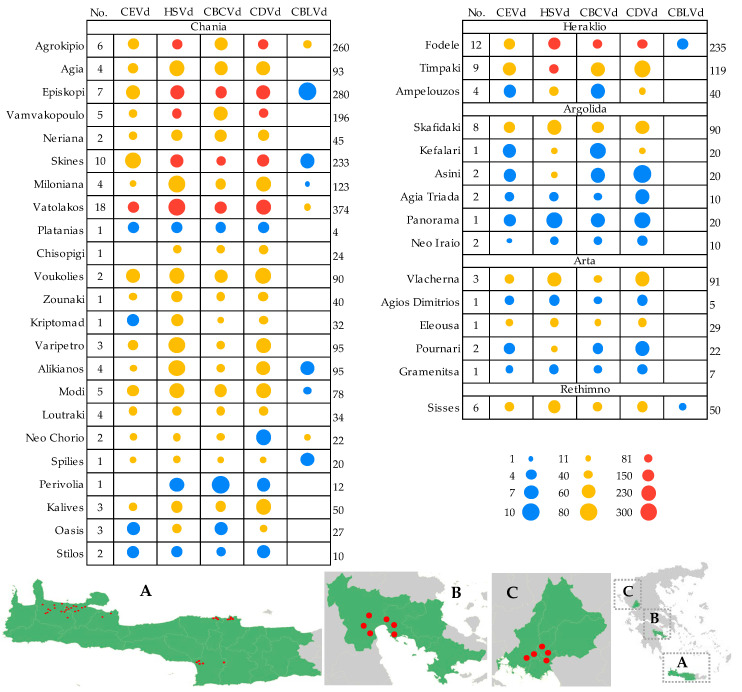
Abundance of total number of positive samples of citrus viroids detected from five provinces, 38 locations (indicated at the left side of table) and 145 different fields (from each location the total number from all fields is presented in the first column (No)). A total of four different scales and three different group colors are presented, in which the size of circles is proportional to the number of infected samples. The total number of collected samples from all fields of each location is shown on the right side of the table. Lower panel: The last right panel is the map of Greece, and the surveyed provinces of Crete (**A**), Arta (**B**) and Argolida (**C**) are indicated in green, and in magnification. The red spots in the magnification pictures indicate the surveyed locations.

**Figure 2 viruses-15-00605-f002:**
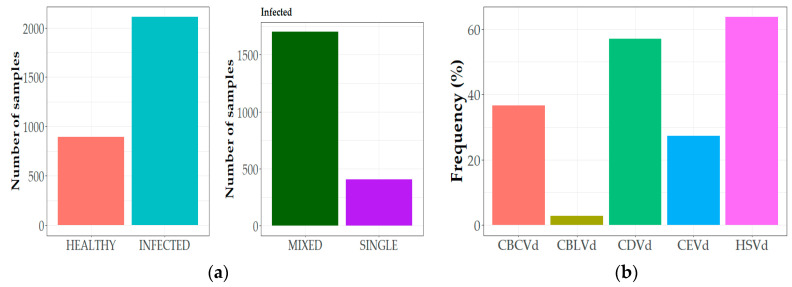
Total number of samples surveyed from five districts in Greece which were found to be healthy or infected, either in single or mixed infections (**a**). Total frequency (%) of the five viroids in Greece (percentage of infected samples in the overall collected samples) (**b**).

**Figure 3 viruses-15-00605-f003:**
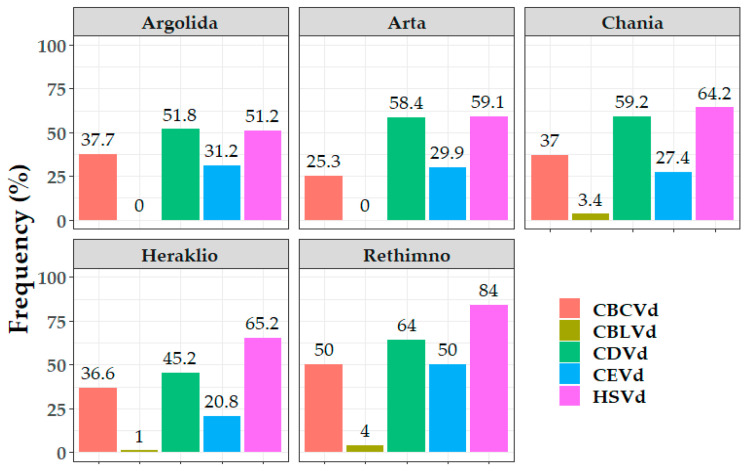
Frequency (%) of citrus viroids in the overall number of samples collected from each geographical district of Greece.

**Figure 4 viruses-15-00605-f004:**
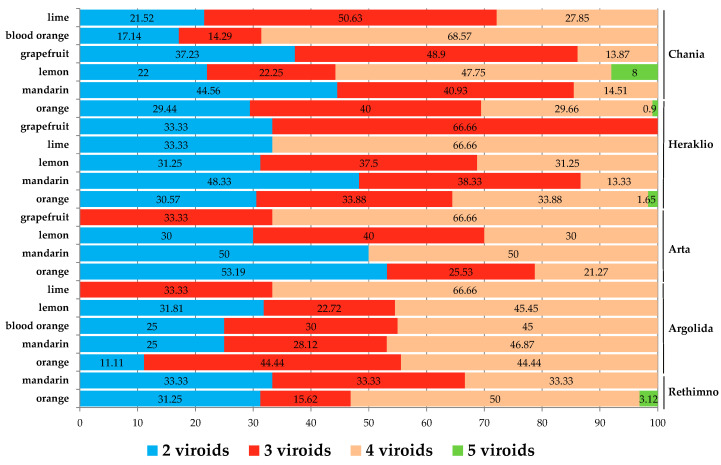
Frequency results (%) of the different mixed infection types (with two, three, four or five viroids) in each surveyed host population in the five districts in Greece.

**Figure 5 viruses-15-00605-f005:**
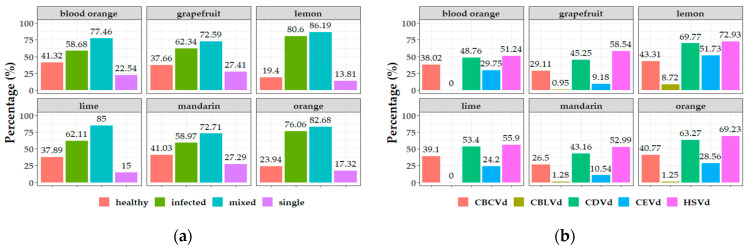
Phytosanitary status (healthy, infected, single, and mixed infected) in percentages (%) of the overall samples of each host collected from five districts in Greece (**a**). Frequency in percentages (%) of citrus viroids in six different hosts from the overall number of collected samples in Greece (**b**).

**Figure 6 viruses-15-00605-f006:**
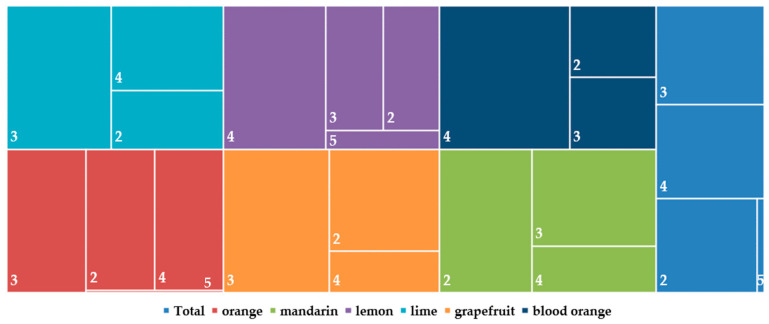
Tree map representation of the mixed infection types in the citrus hosts which were evaluated for the presence of five viroids. The colors of the boxes reflect the six specific surveyed hosts and the total sum of the samples. The numbers of the detected viroids (mixed infection type) are given inside boxes (2 = infection with two viroids, 3 = infection with three viroids, etc). The size of the boxes is proportional to the percentage of the infection types in each category.

**Table 1 viruses-15-00605-t001:** Results of the conducted survey for frequency evaluation of citrus viroids in Greece. Results are presented as percentages (%) of the infected samples per host cultivar in each geographical district.

**CHANIA**	**CEVd**	**HSVd**	**CBCVd**	**CDVd**	**CBLVd**	**ARTA**	**CEVd**	**HSVd**	**CBCVd**	**CDVd**	**CBLVd**	**HERAKLIO**	**CEVd**	**HSVd**	**CBCVd**	**CDVd**	**CBLVd**
Valencia	18.18	66.06	38.18	60.61	3.03	Lane Late	10	70	20	40	0	Valencia	0	55	35	30	0
Chalder	22.5	67.5	25	72.5	0	W. Navel	28.57	42.86	28.57	42.86	0	New Hall	18	76	24	44	0
Lane Late	30.16	65.87	36.51	61.9	0	Mpotsato	12.5	67.5	10	75	0	W. Navel	20	53.33	43.33	43.33	6.67
New Hall	33	72	45	72	0	Saloustiana	52.63	63.16	52.63	68.42	0	Mirodato	36.96	71.74	52.17	61.96	0
W. Navel	29.81	78.26	49.69	74.53	0	Adamopoulou	60	80	40	80	0	Soultani	55	75	50	60	0
N. Late	40	67.5	42.5	60	10	Nuvel Athous	51.72	68.97	48.28	72.41	0	Zampetaki	46.67	86.67	33.33	53.33	0
Xino Chanion	34.55	67.27	47.27	63.64	0	Red	60	60	40	60	0	Adamopoulou	42.86	71.43	0	42.86	0
Zampetaki	42.86	69.29	35.71	67.14	7.14	Nova	0	0	0	33.33	0	Eureka	75	50	50	50	0
Adamopoulou	61.07	74.81	55.73	74.81	9.16	Clementine	35	65	15	50	0	White	0	80	60	40	0
Vakalou	65.69	81.75	55.47	78.1	20.44	Moro	0	28.57	0	0	0	Ancor	0	70.97	45.16	71.94	6.45
Interdonato	32.5	71.25	23.75	68.75	10	Persian	0	0	0	0	0	Clementine	16	52	28	32	0
Eureka	51.95	72.73	44.16	67.53	0							Kino	3.3	53.85	23.08	31.87	0
Nova	0	31	6	42	0	**ARGOLIDA**	**CEVd**	**HSVd**	**CBCVd**	**CDVd**	**CBLVd**	Persian	50	75	50	75	0
Ancor	7.95	50.99	18.54	40.4	4.64	Valencia	20	60	60	40	0						
Clementine	23.88	56.72	43.28	49.25	0	Lane Late	20	20	0	60	0	**RETHIMNO**	**CEVd**	**HSVd**	**CBCVd**	**CDVd**	**CBLVd**
Kino	0	55.74	52.46	54.1	0	New Hall	0	0	0	0	0	New Hall	53.57	85.71	57.14	67.86	7.14
Page	12.5	67.5	21.25	43.75	1.53	W. Navel	30	60	50	70	0	W. Navel	50	83.33	41.67	75	0
White	8.33	45.83	13.33	36.67	0	Interdonato	42.5	52.5	32.5	50	0	Nova	40	80	40	40	0
Red	11.76	77.21	51.47	63.24	2.21	Tarocco	30	60	50	70	0						
Pink	0	36	2	16	0	Moro	25	60	60	60	0						
Tarocco	25.81	51.61	25.81	43.89	0	Nova	50	52.5	35	50	0						
Moro	37.04	48.15	51.85	44.44	0	Clementine	50	55	40	60	0						
Kara Kara	43.75	75	56.25	75	0	Page	20	40	35	40	0						
Persian	23.97	56.85	39.73	54.79	0	Persian	40	80	60	60	0						

## Data Availability

Not applicable.

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
