# Peer review of "Incidence and Epidemiology of Citrus Viroids in Greece: Role of Host and Cultivar in Epidemiological Characteristics"

_viruses, 2023, doi:10.3390/v15030605_

Round 1
Reviewer 1 Report
The manuscript Incidence and Epidemiology of Citrus Viroids in Greece: Role of Host and Cultivar in Epidemiological Characteristics by Mathioudakis et al., reported the epidemiology of different citrus viroids in different citrus in Greece that provide a deep and extensive investigation of citrus viroids in Greece area.
In this manuscript, the data they collected was abundant and analyzed comprehensively, and also provided some new knowledge about the citrus species-viroid correlation.
There are some issues pointed as followed:
- In the fourth paragraph of the introduction, authors should indicate the eight viroids(CEVd, CDVd, CBCVd, CBLVd, HSVd, CDd-V, CDd-VI, CDd-VII) belong to which genera.
- In figure1, the number dots labeled by yellow and green color are not easy to distinguish in size, making the green and yellow dots similar to the blue ones will be better.
- Since the data were collected in four periods, two autumn and two spring, I suggest analyzing the data according to these two different periods will provide more information about the season affection on host-viroids correlation.
- In figure 3, the prevalence of CDVd and HSVd are 51.8 and 51.2, I don’t think there exists significance between those two numbers, so cannot be said “CDVd was prevalent”, they are peer perform.
- In figure 6, in the orange part, squares of infection with 4 viroids and 5 viroids are not drawing well. Including the infection of one viroid in the tree map will give readers more intuitive data.
- In the discussion part, should indicate the geography and climate differences in the three surveyed regions and whether these elements affect the citrus-viroids correlation.
Author Response
The manuscript reported the epidemiology of different citrus viroids in different citrus in Greece that provide a deep and extensive investigation of citrus viroids in Greece area.
In this manuscript, the data they collected was abundant and analyzed comprehensively, and also provided some new knowledge about the citrus species-viroid correlation.
Answer: We thank the Reviewer#1 for the positive and constructive criticism.
- In the fourth paragraph of the introduction, authors should indicate the eight viroids (CEVd, CDVd, CBCVd, CBLVd, HSVd, CDd-V, CDd-VI, CDd-VII) belong to which genera.
Answer: The information about citrus viroids genus are now incorporated in the Introduction section as: “To date, eight viroid species assigned to four different genera of the family Pospiviroidae infecting citrus have been reported: citrus exocortis viroid (CEVd) of the genus Pospiviroid, hop stunt viroid (HSVd) of Hostuviroid, citrus bark cracking viroid (CBCVd) of Cocadviroid, and citrus dwarfing viroid (CDVd), citrus bent leaf viroid (CBLVd), citrus viroid V (CVd-V), citrus viroid VI (CVd-VI), citrus viroid VII (CVd-VII) of Apscaviroid.”
- In figure1, the number dots labeled by yellow and green color are not easy to distinguish in size, making the green and yellow dots similar to the blue ones will be better.
Answer: The Figure 1 is revised as suggested by incorporating three instead of four color groups in order the labeled dots scale size between groups (automatically crated by the software) to be similar and easier to distinguish. In the Figure.1 the legend was revised as “Four different scales and three different group colors are presented in which the size of circles is proportional to the number of infected samples”. Moreover, additional information is given in Materials & Methods section (lines 142-145).
- Since the data were collected in four periods, two autumn and two spring, I suggest analyzing the data according to these two different periods will provide more information about the season affection on host-viroids correlation.
Answer: We thank the reviewer for this comment. The aim of this work was to assess the incidence and epidemiology of citrus viroids and to address this, samples were collected only once and all were from different fields and locations of the five geographic areas. Nevertheless, we have analyzed those data according to these two periods (autumn and spring) and the viroids frequency rates were robust and similar between them. Thus, we have added this sentence in the text (lines 158-160): “Similar frequency rates of infected samples were observed when we analyze the total number of sample hosts and pathogens in each collection period (autumn, spring), confirming, as already known, the suitability of both periods for citrus viroids detection”. Repetitive sampling of the same trees in different seasons is something we are planning for a future work based on the data collected in this present study (to focus at specific plots in some locations of an area and test the same samples in different periods).
- In figure 3, the prevalence of CDVd and HSVd are 51.8 and 51.2, I don’t think there exists significance between those two numbers, so cannot be said “CDVd was prevalent”, they are peer perform.
Answer: Revised as suggested: “HSVd was the most predominant between 51.2 and 84% in all districts followed by CDVd (45.2-64%), except Argolida district in which both were peer perform detected (Figure 3)”.
- In figure 6, in the orange part, squares of infection with 4 viroids and 5 viroids are not drawing well. Including the infection of one viroid in the tree map will give readers more intuitive data.
Answer: In orange trees, the infection rate with 5 viroids was low (1.07%) compared to lemon trees (6.93%) or the total sum of sampled hosts (2.29%). This low number in orange trees made difficult the square to be more visible compared to the other two (as the size of each square is proportional to the percentages of infected samples with each mixed type of infection). The software automatically drawn and arranged how the squares will be presented (the 5-viroids square was placed under the squares of 2- and 4-viroids) and unfortunately, we cannot interfere to change it, but still when you zoom the figure it is clear. We managed only to move the number 5 closer to its square to be more distinguishable.
Regarding the second scale of this comment, although as the Reviewer#1 states correctly that the inclusion of single infections in this tree map would give the overall data to the reader, currently this cannot be done as these results were already mentioned and presented in the text and Figure.5 (lines 257-260, Figure 5a) for each host. Therefore, this would be a repetition in figures presentation, and mainly because the Figure.6 focuses in the data presentation of a different infection type (mixed infections; Figure.5a was for single infection type) and evaluation of potential correlation of this parameter with the hosts, the inclusion of single infections will change the ratios presented.
- In the discussion part, should indicate the geography and climate differences in the three surveyed regions and whether these elements affect the citrus-viroids correlation.
Answer: We have added these sentences in the discussion as suggested (lines 343-348): “The surveyed areas are covered by lowlands. These areas have similar geomorphological/landscape characteristics and are dominated by non-sloping farms. Regarding the climate conditions, there are some differences between southern to middle-northwestern studied areas, which are mainly noted in the amount of annual rainfall, while minor differences are observed in the range of average temperatures. The sampling time in the two northern areas was adjusted to coincide with similar temperature conditions of the southern areas and achieve overall uniformity among the studied areas.”

Reviewer 2 Report
English language is unclear in many places throughout the text. Please proofread.
Materials & Methods are not sufficiently described.
I do not see statistical analysis throughout the data figures or tables.
The huge gap in sample size needs to be statistically justified.
I think frequency is a better term to be used in replacement of prevalence (such in figure 3).
Tables are very hard to read or interpret. No adequate explanation is provided. For example, Table S4: it is unclear how this data has been calculated. Data in table S4 refers to single infections while the sum of each viroid-infected samples is more than the number of total samples.
Author Response
Comments from Reviewer #2
- English language is unclear in many places throughout the text. Please proofread.
Answer: As suggested a proofreading by a native English speaker has been undertaken.
- Materials & Methods are not sufficiently described.
Answer: We have revised this part and included additional text at section 2.4 (lines 143-152), describing in detail the processing of data and how analysis was performed with examples. Some other details have been also included (lines 85, 91, 103-104, 110-111, 129-130).
- I do not see statistical analysis throughout the data figures or tables.
Answer: The information regarding the data statistical processing that was carried out is now added in Materials and Methods section 2.4 and in Suppl. Table S7. The obtained data cannot be processed by statistical software as they come from field surveys and their presentation is deterministic (the samples were only collected once, not repetitive sampling of same trees and same plots during the periods to have samples replications) and not stochastic which follows analysis by statistical software. In fact, we tried as suggested to use the data of each field from an area as “potential” replications and when we performed the ANOVA test the percentages changed due to the fact that it calculates the average percentages of infection from each field leading to different ratios as the small scale sampled fields give higher values to the average (eg a field of 120 trees with 30 infected trees has 25% infection and a field of 10 trees with 8 infected has 80%. In ANOVA it calculated the average of these two as 52.5% which is contradictory to the real data: from 130 trees the 38 were infected, 27.5%).
- The huge gap in sample size needs to be statistically justified.
Answer: Revised and incorporated in lines 101-103. In fact, some of the species were either not possible to be found in some districts (due to climate conditions or soils demands are not cultivated) or were surveyed in a lower number due to limited available fields. Additional information was given in introduction and discussion (lines 49-51, 318-320) reporting that oranges and mandarins are the most cultivated species in Greece and by default their sampled size was higher compared to less cultivated species (present in specific areas).
- I think frequency is a better term to be used in replacement of prevalence (such in figure 3).
Answer: As suggested the term “prevalence” has been replaced by “frequency” in text and figures.
- Tables are very hard to read or interpret. No adequate explanation is provided. For example, Table S4: it is unclear how this data has been calculated. Data in table S4 refers to single infections while the sum of each viroid-infected samples is more than the number of total samples.
Answer: We thank the reviewer for his comment. Table legends are revised with more information. Indeed, the amount of data is huge and following the Reviewer#2 suggestion we added a Suppl. Table S7 in which all the adequate explanations and formulas used for the data statistical processing are presented. Regarding the Table S4: the following formulas were used
F infected = (number of infected samples by each viroid per host in each area) / (number of collected samples per host in each area) x 100
F single infections = (number of single infected samples by each viroid per host in each area) / (number of infected samples per host population in each area) x 100
Please note that Table S4 is separated in two sub-tables. In one hand, the left sub-table shows the total number of single infections from all hosts in each area, the total number of single infected samples from all hosts for each viroid species, the number of single infected samples for each viroid per host, the number of infected samples per host in parenthesis and the percentage [Example: in Rethimno we had 4 single infected samples with HSVd, 2 from oranges and 2 from mandarins. The frequency of single infections for HSVd in oranges is 2/34 (number of infected orange samples) = 5.88%].
In the other hand, the right sub-table shows the total number of infected samples (mixed infections plus single infections) from all hosts in each area, the number of collected samples per host in each area, and for each viroid the number of infected samples per host is given with the frequency in parenthesis. [Example: in Rethimno we had 21 infected orange samples with CEVd, and the percentage is 21/40 (number of collected orange samples) = 52.5%].
For each of the figures and tables we provided the formulas used in analysis with examples as the Suppl. Table 7.

Round 2
Reviewer 2 Report
Please revisit my previous comments.
This work is precious, and the data will add value to the science. However, I think the data need to be represented in a different way, and the manuscript need to be rewritten.
Author Response
1. There are quite many language/writing style issues; below I list a few:
L18, “that lead their frequency”, consider using “leading to their occurrence”.
Answer: The manuscript was proofread by native speaker and English expressions in some parts were accordingly revised.
L18: the change was incorporated in the text as suggested (“leading to their occurrence”).
2. L20, “six citrus species of 29 cultivars”, consider using “29 cultivars of six citrus species”.
Answer: the change was incorporated in the text as suggested (“29 cultivars of six citrus species”).
3. L21, “the frequency of…”, consider using “the occurrence of”.
Answer: the change was incorporated in the text as suggested (“the occurrence of”).
4. L40, “as citrus play a staple crop of a rich source for nutrients and antioxidants”, this sentence is awkward and contains misinformation. Is citrus classified as staple crop? I don’t think so. Consider rephrase to something simple, such as “as citrus fruits provide a rich source for nutrients and antioxidants”
Answer: the change was incorporated in the text as suggested (“as citrus fruits provide a rich source for nutrients and antioxidants”).
5. L42, “especially those,”, delete the comma.
Answer: the change was incorporated in the text as suggested.
- L43, “performance factors” is ambiguous.
Answer: the following replacement was incorporated in the text: “degrading tree yield factors and final product quality”.
